# Physics-Transfer Learning: A Framework to Address the Accuracy-Performance Dilemma in Modeling Complexity Problems in Engineering Sciences

## Abstract

The development of theoretical sciences traditionally adheres to an observation-assumption-model paradigm, which is effective in simple systems but challenged by the 'curse of complexity' in modern engineering sciences. Advancements in artificial intelligence (AI) and machine learning (ML) offer a data-driven alternative, capable of interpolating and extrapolating scientific inference where direct solutions are intractable. Moreover, feature engineering in ML resembles dimensional analysis in classical physics, suggesting that data-driven ML methods could potentially extract new physics behind complex data. Here we propose a physics-transfer (PT) learning framework to learn physics across digital models of varying fidelities and complexities, which addresses the accuracy-performance dilemma in understanding representative multiscale problems. The capability of our approach is showcased through screening metallic alloys by their strengths and predicting the morphological development of brains. The physics of crystal plasticity is learned from low-fidelity molecular dynamics simulation and the model is then fed by material parameters from high-fidelity, electronic structures level, density functional theory calculations, offering chemically accurate strength predictions with several orders lower computational costs. The physics of bifurcation in the evolution of brain morphologies is learned from simple sphere and ellipsoid models and then applied to predict the morphological development of human brains, showing excellent agreement with longitudinal magnetic resonance imaging (MRI) data. The learned latent variables are shown to be highly relevant to uncovered physical descriptors, explaining the effectiveness of the PT framework, which holds great potential in closing the gaps in understanding complexity problems in engineering sciences.

## 1 Introduction

The development of theoretical frameworks in engineering sciences has traditionally adhered to an observation-assumption-model paradigm, exemplified by Galileo's studies on beam bending to the formulation of dislocation theory in the mechanical behaviors of materials. This method is particularly effective in problems with a low-dimensional parameter space, where the complexity can often be captured by analytical models. However, as we expand into the multiscale understanding of matter, the 'curse of complexity' emerges, making it increasingly challenging to capture the intricate physics with purely analytical methods (Fish et al., 2021). For instance, material strength is governed by phenomena across multiple length and time scales. Even for single crystals, dislocations can be nucleated under mechanical loads, evolving cooperatively into complex networks (Oh et al., 2009). Brain development involves gene expression, cellular behaviors, and mechanical instabilities across various spatiotemporal scales, as reflected in the evolving morphologies (Llinares-Benadero & Borrell, 2019a). First-principles theories offer high accuracy but are challenging to scale. Empirical models, while highly efficient, are constrained by the limitations of their assumptions and uncertainties in parameterization. This is the accuracy-performance dilemma in modeling complexities of multiscale physics in engineering sciences.

Recent advancements in machine learning (ML) and artificial intelligence (AI) present a promising, data-driven alternative (Fig. 1a). This emerging approach, while still constrained by the density and coverage of data, offers an increasingly powerful tool as data quality and quantity improve. The ability of ML models to interpolate and extrapolate improves accordingly, suggesting that these tools can complement traditional theories where direct solutions become impractical or intractable (Zhang et al., 2018; Li et al., 2022). Moreover, the process of feature engineering in ML bears a resemblance to dimensional analysis in classical physics, offering a systematic way to uncover and utilize internal correlations within complex data (Xu et al., 2022b). This parallel suggests that ML, through its data-driven methods, could potentially extract and transfer physical insights across digital models of varying fidelity and complexity.

Inspired by these thoughts, we propose a physics-transfer (PT) framework to learn physics across digital models of varying fidelities and complexities (Fig. 1b). The learned physics is used for scientific inference with high accuracy and performance to address the dilemma in modeling the complexity. Two representative cases are chosen to demonstrate the capabilities of PT framework including materials strength screening and predicting the development of brain morphologies, which encompass inorganic matter and organs and involve multiscale physics (Fig. 1c). The physics of crystal plasticity is initially learned through low-fidelity molecular dynamics simulations, and these insights are subsequently utilized in high-fidelity density functional theory computations of material parameters, enabling chemically accurate strength predictions. The physics of bifurcation in brain morphologies is initially learned using spherical models with simple geometries and then applied to predict the evolutional behaviors of human brains. The proposed framework holds great potential for enhancing our comprehension of complexity problems in engineering sciences and bridging gaps between understandings from modeling and experimental data.

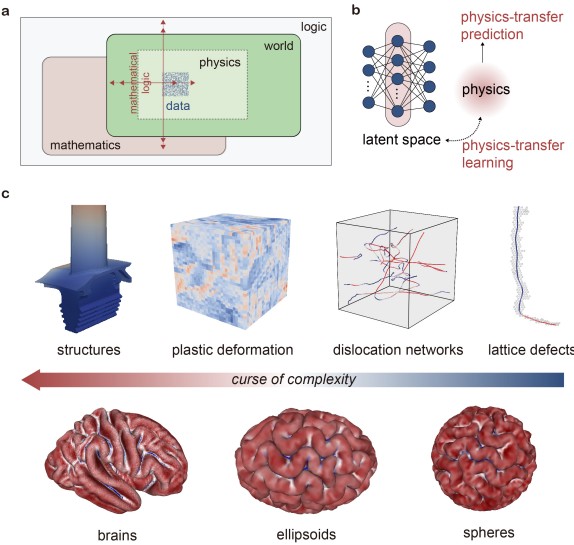

Figure 1: Accuracy-performance dilemma in modeling multiscale physics and proposed physics-transfer (PT) learning framework. **(a)** Machine learning, constrained by data density and coverage, serves as a potent complement to traditional theories for interpolating and extrapolating solutions, especially as data quality and quantity increase. **(b)** The PT learning framework learns physics across digital models of varying fidelity and complexity, enabling extrapolation to effectively address the accuracy-performance dilemma. **(c)** The 'curse of complexity' in multiscale physics of inorganic matter and organs.

## 2 PHYSICS-TRANSFER LEARNING FRAMEWORK

Models with different fidelity ($\mathcal{F}$) in multi-scale modeling exhibit distinct parameters distributions $p(\theta|\mathcal{F})$, where $\theta$ are model parameters, and $p(\cdot|\cdot)$ is conditional probability. ML and AI models

can provide general model with parameters distributions $p(\theta|\mathcal{D})$ based on data ($\mathcal{D}$). Typically, data with different fidelity (*e.g.*, low fidelity ($\mathcal{D}_{LF}$) in molecular dynamics (MD), high fidelity ($\mathcal{D}_{HF}$) in density functional theory (DFT)) will result in ML models having different parameter distributions, that is:

$$p(\theta|\mathcal{D}_{LF}) \neq p(\theta|\mathcal{D}_{HF}), \tag{1}$$

which limits the transferability and extrapolation of models trained on data with different fidelity.

The physics ($\mathcal{P}$) behind the $\mathcal{D}$ can assist the extrapolation with a 'physics-transfer' paradigm. Specifically, if there is a physical relationship between features ($\mathbf{x}$) and target ($\mathcal{O}$) in $\mathcal{D}$, that is:

$$\mathbf{x} \xrightarrow{\mathcal{P}} \mathcal{O}, \tag{2}$$

$$\mathbf{x} \cap \mathcal{O} = \mathcal{D}^{'} \subset \mathcal{D}, \tag{3}$$

the designed ML models ($h \in \mathcal{H}$) can learned the physics behind the data, and models trained on data with different fidelity would have similar parameter distributions, that is:

$$p(\theta|\mathcal{D}^{'}_{LF}) \approx p(\theta|\mathcal{D}^{'}_{HF}), \tag{4}$$

which makes the transferability and extrapolation of models with different fidelity possible and bridge gaps between different modeling methods in multi-scale modeling. To validate the effectiveness of the PT learning framework, in the next sections, we train models on low-fidelity or simple-geometry data and then perform zero-shot extrapolation directly to high-fidelity and high-complexity data. We assess the accuracy of the prediction results and demonstrate its role in addressing the accuracy-performance dilemma.

## 3 EXPERIMENTS

To demonstrate the effectiveness of the PT learning framework, we perform experiments on the problems of materials strength screening and predicting the development of brain morphologies. These two issues encompass the multiscale complexity of inorganic matter and organs suffering from the accuracy-performance.

### 3.1 PHYSICS-TRANSFER LEARNING FOR MATERIALS STRENGTH SCREENING

The strength of materials, like many problems in the natural sciences, spans multiple length and time scales, and the solution has to balance accuracy and performance. In the crystal plasticity (CP) theory, plastic flow and hardening behaviors during material deformation are modeled in a multiscale framework bridging the atomic-scale lattice dynamics and continuum-level stress/strain fields (Roters et al., 2011). One of the key material parameters in CP models is the critical resolved shear stress (CRSS), $\tau_c$, which determines the activation of specific slip systems. In CP models, CRSS is a phenomenological parameter often obtained by fitting experimental results (Salem et al., 2005; Gong et al., 2015). Alternatively, the Peierls stress ($\tau_P$) defined as the minimum shear stress required to move a single dislocation of unit length in a perfect crystal in the absence of thermal activation is also used in the literature for CRSS (Shimanek et al., 2022). The Peierls stress can be obtained from full-atom MD simulations. However, the strain inhomogeneity induced by a dislocation usually spans $10 - 20$ nm, which cannot be directly calculated from first-principles calculations. Previous studies are thus limited by the use of empirical force fields (Soleymani et al., 2014). In practice, the Peierls-Nabarro (PN) model offers a simplified and approximating approach to derive the Peierls stress with the assumptions of sinusoidal interfacial restoring stress and a rigidly shifting dislocation, where the structure of a dislocation core is determined by minimizing the elastic energies and lattice misfit (Nabarro, 1947). The success of the PN model suggests that the Peierls stress is controlled by the elastic responses of the crystals and the energy landscape of interfacial slips (Bulatov & Kaxiras, 1997; Nabarro, 1997; Lu et al., 2000; Rodney et al., 2017).

By assuming the existence of such correspondence, we use the PT learning framework to predict the Peierls stress for a wide spectrum of metallic alloys and inorganic crystals at the first-principles level (Fig. 2a). The maps between the Peierls stress ($\mathcal{O}$) and characteristic materials parameters ($\mathbf{x}$) are trained from empirical or machine-learning force-field (MLFFs) MD simulations with a designed neural network ($h \in \mathcal{H}$) and obtain the posterior probability of the model parameters ($p(\theta|\mathcal{D}^{'}_{LF})$).

Then the well-trained models are extrapolated to DFT-calculated parameters to make predictions at the chemically accurate level. This mapping transfers the physics from low-fidelity but efficient force field models to the first-principles methods, successfully resolving the accuracy-performance dilemma.

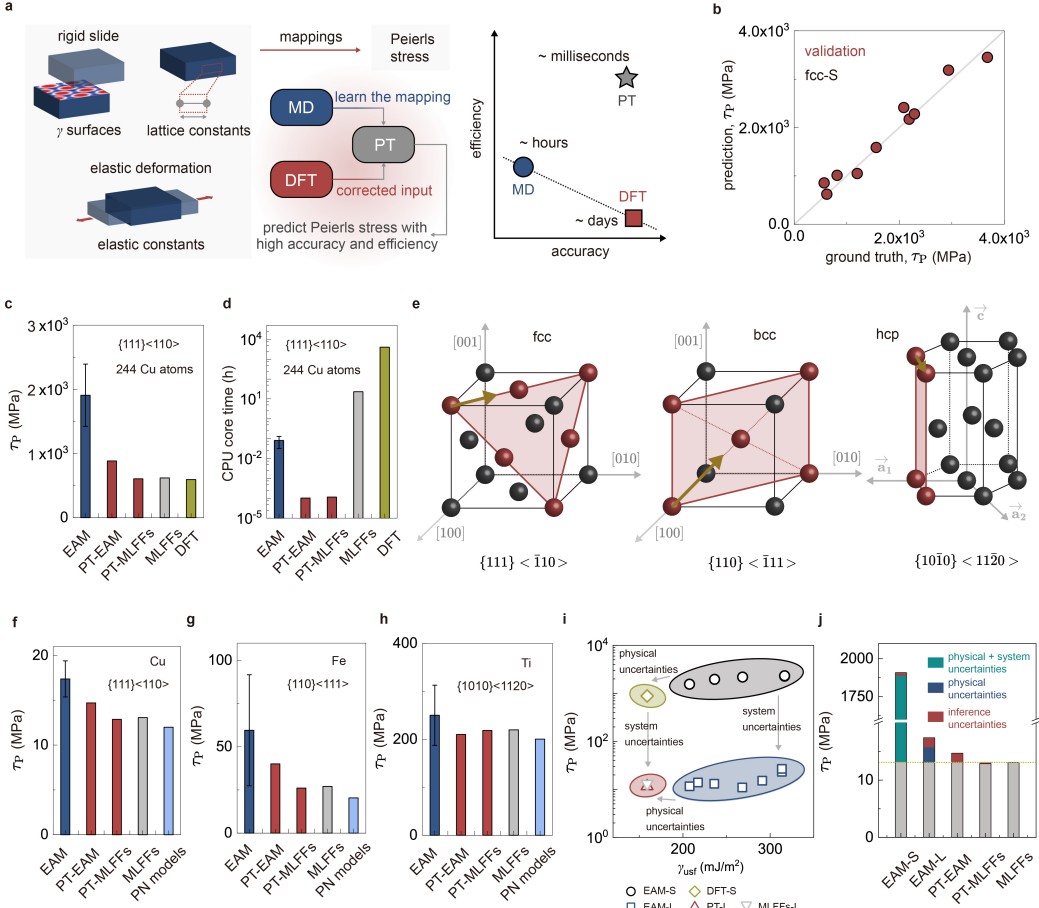

Figure 2: PT predictions for materials strength and uncertainty quantification. **(a)** PT framework transfers the physics from low-fidelity force field models to chemically accurate first-principles methods, effectively addressing the trade-off between accuracy and computational expense. **(b)** Well-trained neural networks learn the physical mapping between the Peierls stress and characteristic materials parameters obtained from atomistic simulation datasets using empirical force fields. **(c, d)** PT framework predicts the Peierls stress with high accuracy and efficiency. The PT predictions are closely aligned with the outcomes of density functional theory (DFT) and machine-learning force-field (MLFF) calculations, with a difference below $48.91\%$, while the results obtained using embedded atom methods (EAM) models deviate substantially from the DFT predictions, with a discrepancy of $221.27\%$ **(c)**. The PT approach also reduces the computational time notably by statistical inference, in comparison with atomistic simulations using DFT, MLFFs, or EAM **(d)**. **(e-h)** PT predictions for different slip systems **(e)**. The PT predictions show good consistency compared to MLFF simulation results (with errors $e = 12.55\%, 48.09\%, 4.30\%$ for Cu $\{111\}\langle\bar{1}10\rangle$, Fe $\{110\}\langle\bar{1}11\rangle$, Ti $\{10\bar{1}0\}\langle11\bar{2}0\rangle$ in prediction, respectively), and superior accuracy compared to EAM ($e = 33.07\%, 72.02\%, 13.89\%$ (**(f),(g),(h)**), respectively). **(i)** Uncertainty quantification shows that the PT predictions eliminate physical and system uncertainties. 'L' denotes the large-supercell system with $\sim 0.8 \times 10^6$ atoms ($160\,\text{nm} \times 2\,\text{nm} \times 40\,\text{nm}$). **(j)** Uncertainty decomposition shows that the inference errors are smaller compared to the physical and system uncertainties. The standard deviation is reported in the error bars.

### 3.1.1 Datasets

To construct the digital libraries, a wide spectrum of metals with crystalline structures of fcc (Cu, Ni, Al, Au, Pd, Pt), bcc (Fe, Mo, Ta, W), and hcp (Ti, Mg, Zr, Co) is explored. The elastic constants, $\gamma$ surfaces, and the Peierls stress are calculated using empirical force fields such as EAM and modified EAM (MEAM) with parameters reported from different sources (Becker et al., 2013; Hale et al., 2018), as well as the lattice mismatch energy and slip resistance. The primary slip systems of fcc ($\{111\}\langle\overline{1}10\rangle$) and bcc ($\{110\}\langle111\rangle$), and the prismatic slip systems of hcp ($\{10\overline{1}0\}\langle11\overline{2}0\rangle$) are considered. Finally, the digital libraries composed of characteristic materials parameters and the Peierls stress are established to learn the physics of crystal plasticity.

### 3.1.2 Architecture and Model Setup

To effectively learn the physical mapping between elastic constants, $\gamma$ surfaces, and the Peierls stress, we employ a convolutional neural network (CNN) to extract features from the $\gamma$ surfaces (He et al., 2016), an feedforward neural network (FNN) to extract the features related to elastic properties and merge them in the latent features space. Then, we use another FNN to predict the Peierls stress. The FNN for extracting elastic properties contains two layers with neuron numbers of 6 (number of elastic features), and 32 (dimension of extracted features), respectively. Following that, the FNN for predicting the Peierls stress has 3 layers with neuron numbers of 64, 32, and 1, respectively. We use the stochastic gradient descent (SGD) optimizer with learning rates of $10^{-4}$ (Hardt et al., 2016).

### 3.1.3 Evaluation Metrics

Recent progress in computational hardware and software promotes the development of MLFFs, which harness neural networks to model the potential energy surfaces (PES) with the precision of the training set, mostly from first-principles calculations (Ko & Ong, 2023; Hedman et al., 2023; Gong et al., 2023). MLFFs learn the dependence of the potential energy of a system on the atomic positions. The size effects in direct DFT calculations can be mitigated if this mapping accommodates all atomic environments encountered in the MLFF simulations, and the locality holds well (Zhang et al., 2018). The Peierls stress predicted by the MLFFs thus serves as a benchmark to validate the PT predictions. However, the accuracy of the state-of-the-art MLFF predictions for non-equilibrium structures such as those containing dislocations is usually limited in comparison with the equilibrium features (Takamoto et al., 2022), and the MD simulations to predict the Peierls stress using MLFFs still need careful design of the models and simulation parameters to mitigate the effects of sample sizes, loading geometries, and kinetics (Morrow et al., 2023). In addition, MLFFs have higher computational costs than common empirical force fields such as EAM and MEAM. A direct mapping between the DFT-derived $\gamma$ surface and the Peierls stress can thus have an advantage in facilitating fast material screening, especially for vast material space.

### 3.1.4 Results

To assess the accuracy of the PT predictions, we first calculate the Peierls stress directly by utilizing different methods of calculations, including EAM, MLFFs, DFT, PT models trained on EAM data (PT-EAM), and PT models trained on MLFFs data (PT-MLFFs) for small systems (annotated as 'S', containing 244 atoms in a $3.48\,\text{nm} \times 0.41\,\text{nm} \times 1.90\,\text{nm}$ supercell) of the fcc system that suffer from strong size effects in predicting the plasticity of bulk materials. The results in Fig. 2b indicate that the well-trained neural networks effectively learn the physical mapping between the Peierls stress and the characteristic elastic and surface parameters. The PT-EAM predictions are quantitatively close ($< 48.91\%$) to those from DFT and MLFF calculations. In comparison, those obtained from EAM models show a significant deviation by $221.27\%$ from the DFT predictions (Fig. 2c for Cu $\{111\}\langle110\rangle$). The time cost of statistical inference in the PT approach is within several milliseconds on an Intel(R) Core(TM) i5-8250U computer with 4 cores), which is significantly lower than that of simulations based on DFT, MLFFs, and EAM (Fig. 2d). These results obtained for small systems successfully demonstrate advantages in the accuracy and efficiency of the PT approach to predict the Peierls stress.

We then consider large models ('L', $\sim 0.8$ million atoms in a $160\,\text{nm} \times 2\,\text{nm} \times 40\,\text{nm}$ supercell) for 3 crystalline structures (fcc, bcc, hcp) with their associated specific slip systems (Fig. 2e), where direct DFT calculations are intractable. MD simulations using MLFFs are performed to validate

the accuracy of PT predictions from EAM and MEAM models. The results show good consistency (with errors $e = 12.55\%$, $48.09\%$, $4.30\%$ for Cu $\{111\}\langle\bar{1}10\rangle$, Fe $\{110\}\langle111\rangle$, Ti $\{10\bar{1}0\}\langle11\bar{2}0\rangle$ in prediction, respectively) and superior performance compared to the EAM results with $e = 33.07\%$, $72.02\%$, $13.89\%$ (Figs. 2f-h). By comparing the results obtained for the small and large systems, we also noted that the size effects are more significant for the empirical force fields. The PT framework thus demonstrates high efficiency compared to DFT and MLFF calculations that can mitigate the size effects, and chemically accurate predictions compared to empirical force fields such as EAM.

For the Peierls stress predictions, uncertainties exist among different theoretical approaches. Uncertainty quantification (UQ) of these methods is of crucial importance in evaluating and selecting the models. Fig. 2i shows the error maps for various calculation methods. The predictions of small systems with EAM (EAM-S) contain physical uncertainties on the potentials and system uncertainties in size effect. The calculations of small systems with DFT (DFT-S) eliminate physical uncertainties but still suffer from system uncertainties. The predictions of large systems using EAM (EAM-L) with weaker size effects reduce system uncertainties but retain the physical uncertainties. Both PT-EAM predictions and MLFFs calculations eliminate physical and system uncertainties, but PT predictions are superior in computational efficiency, in both the training and inference processes. The uncertainties of different approaches are quantitatively decomposed in Fig. 2j. The uncertainty of prediction using EAM-S contains physical, system errors ($99.05\%$ in total) and the inference error ($0.95\%$) by considering the MLFF results as the ground truth. For EAM-L, their contributions are $62.85\%$ and $37.15\%$, respectively. The PT-EAM prediction only involves uncertainty of inference ($e = 12.55\%$). The low uncertainty of inference compared to the physical and system errors demonstrates the power of the PT framework and can be estimated from the theory of machine learning (Abu-Mostafa et al., 2012; Feng et al., 2023).

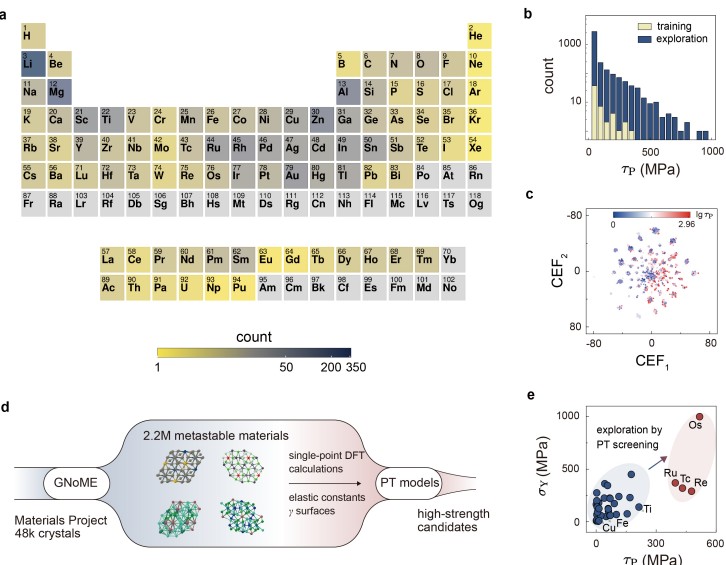

Figure 3: Material strength screening using PT approach. **(a)** The material strength database constructed by PT learning, which covers 88 elements across the periodic table. **(b)** The distribution of $\tau_P$ in the material strength database. **(c)** Distribution of $\tau_P$ in the space of chemical compositions, visualized by t-distributed stochastic neighbor embedding (t-SNE). The crystals are represented by the sum of the one-hot encodings of their constituent elements. The t-SNE reduces the high-dimensional representations of crystals to two principal features (crystal embedding features 1 and 2, $CEF_1$ and $CEF_2$)) (Van der Maaten & Hinton, 2008) **(d)** High-strength material screening from the extensive space of metastable materials in GNoME. **(e)** High-strength materials screened using PT learning and the corresponding yield strengths ($\sigma_Y$) reported in experiments (extracted from MatWeb (Ross, 2013)).

The high accuracy and efficiency of the PT framework allow for single-crystal strength screening and the implementation of mesoscale physics such as the grain texture into the paradigm of high-

throughput materials screening and discovery. For a given material genome including the elements and lattice types, the characteristic materials parameters can be estimated by the equilibrium properties reported in the Materials Project. For example, the $\gamma$ surface can be fitted from a few single-point energy calculations (*e.g.*, intrinsic stacking fault energy (SFE) $\gamma_{\mathrm{isf}}$, unstable SFE $\gamma_{\mathrm{usf}}$, aligned SFE $\gamma_{\mathrm{asf}}$, and the energies of their intermediate configurations) and interpolated using the Fourier series (Su et al., 2019). The elastic constants can be determined by the slope of the linear region in stress-strain curves. The generated characteristic parameters can be used to screen materials by their strengths, a Holly Grail in materials science, through the predicted Peierls stress, and extension by implementing mesoscale physics models such as CP (Roters et al., 2011). Recent advances in theoretical materials science accelerated by artificial intelligence significantly expanded the space of scientific exploration. The graph networks for materials exploration (GNoME) model enlarges the library of inorganic crystals from 48k to 2.2M, many of which are metastable materials that have not been synthesized by existing methods and thus cannot be assessed by experiments (Merchant et al., 2023). The PT model can efficiently screen materials in such a huge library at the chemically accurate level, especially for non-equilibrium material properties and processes inaccessible by conventional approaches due to the accuracy-performance dilemma. Specifically, $3,471$ fcc (Fm$\overline{3}$m), bcc (Im$\overline{3}$m) or hcp (P6$_3$/mmc) crystals of the 2.2M inorganic crystalline compounds in GNoME are supplemented with calculated elastic properties and chosen for material strength screening (Figs. 3a-c). A product material strength database is finally constructed (Figs. 3a-c). High-strength metal materials (Os, Ru, Tc, Re) screened out from the database are verified by their experimentally measured yield strengths ($\sigma_Y$) (Ross, 2013), and are much stronger than the metals in the training set (*e.g.*, fcc Cu, bcc Fe, hcp Ti) (Figs. 3d and 3e).

## 3.2 PHYSICS-TRANSFER LEARNING FOR THE PREDICTION OF BRAIN MORPHOLOGY DEVELOPMENT

Brain development involves complex multiscale physical processes, encompassing gene expression, protein folding, and cellular behaviors such as cell division, differentiation, and migration, as well as macroscopic morphological instabilities (Llinares-Benadero & Borrell, 2019a). The continuum mechanics theory that incorporates growth tensor parameters is widely used to describe the morphological evolution of tissue growth (Tallinen et al., 2016; Striedter et al., 2015; Darayi et al., 2022; Budday & Steinmann, 2018; da Costa Campos et al., 2021; Alenyà et al., 2022). These growth tensor parameters can be linked to micro-scale cellular behaviors, providing a multiscale modeling framework for modeling morphological instabilities. The intricate geometry of the brain and the nonlinear nature of brain morphological development involving materials and contact result in low computational efficiency and poor convergence in finite element analysis (FEA) (Tallinen et al., 2016). Consequently, there is limited work directly simulating the morphological development of the brain, with most studies discussing that on simplified geometries, such as two-dimensional shell-substrates geometries, or three-dimensional spheres and ellipsoids (Fig. 4a) (Darayi et al., 2022; Budday & Steinmann, 2018; da Costa Campos et al., 2021; Wang et al., 2021). Indeed, the growth of spheres or ellipsoids shares similar spatiotemporal characteristics with brain morphological development, such as ridge-valley networks and bifurcation behaviors. By designing neural network architecture ($h \in \mathcal{H}$), one can learn the physics of bifurcation and morphological features from simple geometries. The well-trained models with parameter distributions of ($p(\theta|\mathcal{D}'_{\mathrm{LF}})$) can be directly extrapolated to predict the morphological development of the high-complexity brain.

### 3.2.1 DATASETS

The experimental data of human brain morphologies is rare, especially for individual brain morphologies (Bethlehem et al., 2022; Ciceri et al., 2024). We collect the currently available open-source brain structural magnetic resonance imaging (MRI) atlases from the source (Ciceri et al., 2024). The pipeline involving cortical and sub-cortical volume segmentation and cortical surface extraction is adopted to obtain brain morphologies from MRI data (Makropoulos et al., 2018). The collected experimental data of human brain morphologies are used to validate the effectiveness of PT learning. We construct digital libraries of morphological patterns involving spheres, ellipsoids, and human brains with increasing geometrical complexities. For spheres and ellipsoids with simpler geometries, a representative core-shell model is used (Tallinen et al., 2014; Wang et al., 2021; Xu et al., 2022a; Yin et al., 2008), as implemented to explore the mechanical instability in cortical folding (Tallinen et al., 2016; Striedter et al., 2015; Darayi et al., 2022; Budday & Steinmann,

2018; da Costa Campos et al., 2021; Alenyà et al., 2022). The outer spherical shell represents the cerebral cortex (gray matter), and the inner core for the white matter. The core and shell structures are modeled as modestly compressible hyperelastic Neo-Hookean material with different growth rates (Tallinen et al., 2016). Following the experimental evidence (Fischl & Dale, 2000; Chang et al., 2007; Xu et al., 2010; Dervaux & Amar, 2008; Budday et al., 2015), the cortical thickness ranges from $0.03 - 1.63$ mm according to the abnormal and normal human cerebral cortex measurements and the scale factor (Fischl & Dale, 2000; Chang et al., 2007), and the relative shear modulus ($\mu_{\text{grey}}/\mu_{\text{white}}$) ranges from $0.65 - 1$ (Xu et al., 2010; Dervaux & Amar, 2008; Budday et al., 2015). The tangential growth (TG) model is used to simulate the cellular mechanisms that create the growth stresses and lead to the pattern evolution (Tallinen et al., 2014; 2016; Llinares-Benadero & Borrell, 2019b).

### 3.2.2 ARCHITECTURE AND MODEL SETUP

In FEA, morphological data are meshed into discretized tetrahedral element, which can be represented as graphs, where the nodes of the graph correspond to the vertices of the elements, and the edges of the graph correspond to the edges of the elements. Consequently, graph neural networks (GNN) are suitable for extracting features from morphologies represented as graphs. Specifically, we utilize an encoder-decoder architecture to learn the complexity of morphological development, constraining the model with the 3D coordinates of the morphologies and global feature su ch as gyrification index (Fig. 4b). The input to the model is a graph representation of the morphology, where the node features include positional coordinates and the normal direction. The output is the local feature curvature of the morphology.

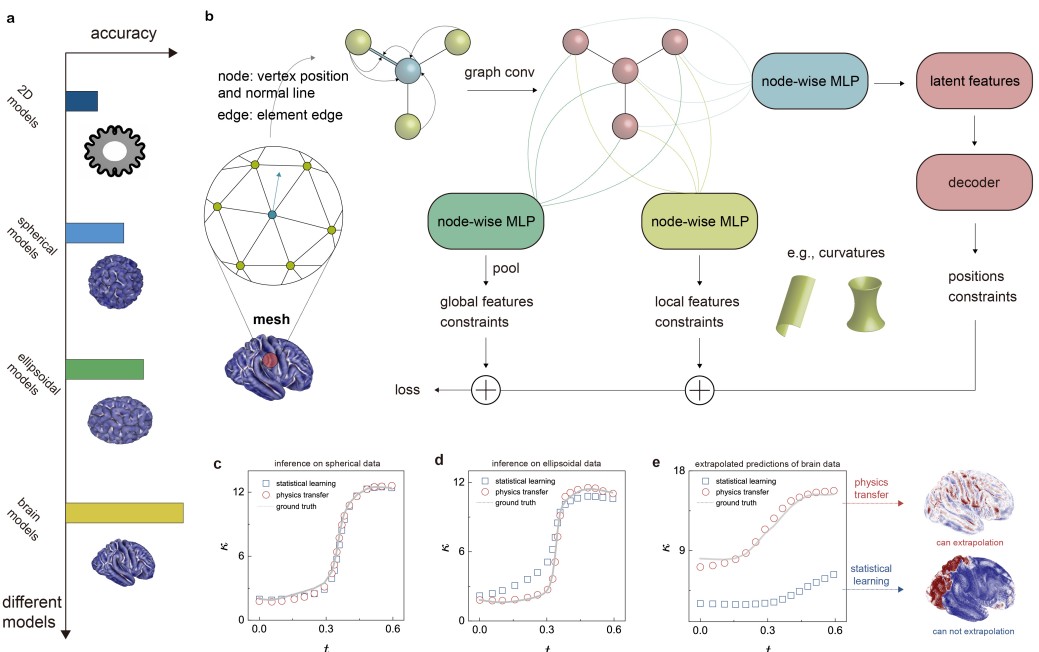

Figure 4: Brain development prediction using PT approach. **(a)** The accuracy of predicting the development of brain morphologies improves with the increase in the geometric complexity of the model. **(b)** An encoder-decoder architecture constrained by multiscale morphological features is used to resolve the morphological complexity. **(c)** The interpolation predictions for spherical data. **(d,e)** The extrapolation predictions for ellipsoidal data **(d)** and the development of brain morphologies **(e)**.

### 3.2.3 Results

We train our models on spherical and ellipsoidal data, which are then applied to the morphological development of human brains. In our ablation study, the features of normal directions are removed and only morphological data is retained, referred to as statistical learning, since the curvatures contain essential physics of the bifurcation processes, which is well known in the nonlinear elasticity community. Our results show that, for the inference of spherical data, both traditional statistical and PT learning yield satisfactory results (Fig. 4c). However, when extrapolating to ellipsoids and human brains using the model trained on the sphere data, PT learning excels while statistical learning fails, highlighting the generalizability of PT learning (Figs. 4d and 4e).

## 4 Related Work

Our PT framework shares some conceptual features with existing ML methods developed to combine multi-fidelity data (Ramakrishnan et al., 2015; Batra et al., 2019; Smith et al., 2019). $\Delta$-learning predicts high-fidelity properties by learning the discrepancies in predictions from models at different levels of fidelity (Ramakrishnan et al., 2015). The objective properties are calculated by correcting low-fidelity calculations following a statistical treatment. In a similar spirit, the low-fidelity as a feature (LFAF) method learns the relation between properties obtained from models with different fidelities, and predicts the high-fidelity properties using objective properties and other materials parameters obtained from low-fidelity models as the input (Batra et al., 2019). Transfer learning pre-trains neural networks on low-fidelity data and fine-tunes the parameters on high-fidelity ones to achieve high accuracy in predictions (Smith et al., 2019). However, these methods are statistical in nature and their applications mainly focus on the properties at equilibrium. In supervised learning, it is necessary and beneficial to label data obtained from high-fidelity models in the training process, which are not available for most non-equilibrium properties such as the Peierls stress with chemical accuracy at the DFT level. Our PT framework resolves this constraint from the accuracy-performance dilemma by going beyond the statistical approach and transferring the physics across models with different fidelities, which is characterized by materials parameters that can be obtained from single-point, unit-cell calculations. For example, the Peierls stress is predicted accurately and efficiently utilizing the learned physics and chemically accurate materials parameters. Longitudinal MRI data of brains are rare, limiting the use of traditional statistical learning methods to directly predict the development of human brain morphologies (Bethlehem et al., 2022; Ciceri et al., 2024). PT learning approach can learn the physics of bifurcation from data of morphological development obtained for simple geometries, and then be applied to the human brain with elevated morphological complexities.

## 5 Discussion and Limitations

Our work 'digitalizes' the observation-assumption-model practice in engineering sciences using neural network representations. As the ML model learns physics from data, physical features naturally emerge in the space of latent variables (Fig. 5). In the case of material strength screening, after the model has learned the physics of crystal plasticity, the principal components of latent features show a weak correlation with the input variables such as the elastic constants (Fig. 5a), but a strong correlation with the dislocation width, another important physical quantity in crystal plasticity (Fig. 5b). In the study of brain morphologies, the ML model of PT learning exhibits similar weight distribution ($p(\theta|\mathcal{D}'_{\mathrm{LF}}) \approx p(\theta|\mathcal{D}'_{\mathrm{HF}})$) after learning from spherical and ellipsoidal data (Fig. 5c), whereas the ML model of statistical learning using the morphology data only shows a significant difference in parameter distribution ($p(\theta|\mathcal{D}_{\mathrm{LF}}) \neq p(\theta|\mathcal{D}_{\mathrm{HF}})$) compared to PT learning (Fig. 5d). The preserved features of weight distribution across data of varying complexity demonstrate the generalizability to complex geometries.

These observations indicate that the PT learning framework captures the essential physics of problems with high complexities, and explains its outstanding performance in addressing the accuracy-performance dilemma. The learned physics in the PT approach is limited by the fidelity of digital libraries, which depend on the completeness of theoretical descriptions and experimental data. Specifically, in materials strength screening, databases constructed with well-trained MLFFs are expected to offer more accurate physics than EAM or MEAM, although their computational costs are

high, and a full set of MLFFs for all metal alloys is not available at present. Our studies show that the error of PT-MLFF predictions using the physics learned from MLFF simulations is reduced to $e = 1.51\%$ (Fig. 2j). This few-shot fine-tuning approach utilizing well-trained MLFFs substantially improves the accuracy of the learned physics compared to the database constructed with EAM potentials (Figs. 2c, 2f-h, 2j). For the prediction of human brain morphologies, the rareness of MRI data could be resolved by the output of ongoing projects such as the Developing Human Connectome Project (dHCP) (Makropoulos et al., 2018) or adding animal data.

The advancement of engineering sciences has often been marked by key moments where fundamental physics is distilled to form theoretical frameworks. Our PT approach continues these efforts by leveraging data representation from real-world problems. By reducing the dimensionality of latent variable spaces and abstracting data correlations, this method has the potential to reveal new theoretical insights, which will be a focus of ongoing research.

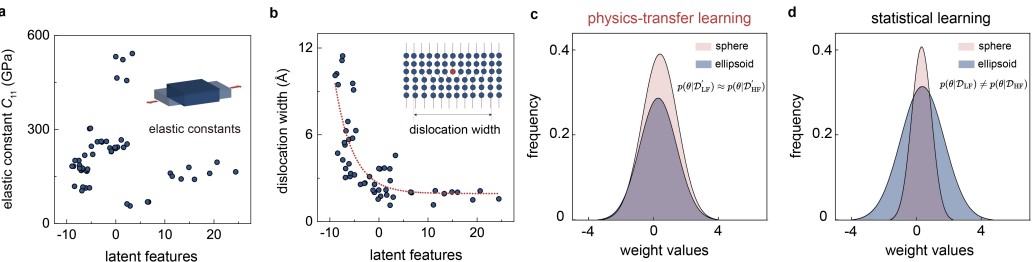

Figure 5: Neural networks analysis for latent space features and weights parameters. **(a,b)** Emergence of physics in the latent space. The model that has learned the physics of crystal plasticity shows low correlation with input variables **(a)**, but high correlation with the key physical variable of dislocation width in crystal plasticity **(b)**. **(c,d)** The weights parameters distribution of ML models trained on the spherical and ellipsoidal data for PT learning **(c)** and statistical learning **(d)**.

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

# A    APPENDIX

## A.1    MD SIMULATIONS

In calculations of the $\gamma$ surfaces, a supercell with lattice vectors of $\mathbf{a}$ ($[1\bar{1}0]$), $\mathbf{b}$ ($[11\bar{2}]$), and $\mathbf{c}$ ($[111]$) for the fcc metal is prepared, which contains 64 atoms and 32 atomic layers along the $z$-axis. A vacuum layer of 30 Å along the $z$-axis is added to avoid interactions between the periodic images of lattice mismatch. The upper 16 atomic layers are rigidly shifted relative to the lower 16 layers progressively along the $z$-axis, and independently in the $x$ and $y$ directions, respectively. Relaxation of the atomic layers along the $z$ direction is allowed after the displacement. The $\gamma$ surfaces are constructed using a $31 \times 31$ grid,

$$\gamma(x, y) = \frac{E_{\mathrm{m}}(x, y) - E_0}{S}, \tag{5}$$

where $E_{\mathrm{m}}(x, y)$ is the energies of the lattice with a mismatch at different displacements, $\mathbf{d} = (x, y)$, $E_0$ is the energy of the crystal in its equilibrium structure, and $S$ is the area of the slip plane.

For the calculations of the Peierls stress, a supercell with $\sim 0.8 \times 10^6$ atoms ($160\,\mathrm{nm} \times 2\,\mathrm{nm} \times 40\,\mathrm{nm}$) is prepared. For the fcc metal, lattice mismatch is created between two half-crystals by shifting along the burgers vector by $\mathbf{a}/\sqrt{2}$. Subsequent structural relaxation then creates an initial edge dislocation (Bulatov & Cai, 2006). Molecular statics calculations are used to calculate the Peierls stress identified as the minimum stress at which the motion dislocation is activated (Lim et al., 2015). A step-wise strain increment of $10^{-5}$ is applied to the supercell. For bcc and hcp metals, similar procedures are adopted but along different lattice orientations and for different slip systems. All MD simulations are performed using the large-scale atomic/molecular massively parallel simulator (LAMMPS) (Plimpton, 1995).

## A.2    DFT CALCULATIONS

To validate the hypothesis and feasibility of the PT framework, we directly calculate the Peierls stress in small systems ('S', with 244 atoms) using Cu as an example. The DFT calculations are carried out using the Vienna *ab initio* simulation package (VASP) using the projector augmented wave (PAW) method and a plane-wave basis (Blöchl, 1994; Kresse & Joubert, 1999). The generalized gradient approximation (GGA) in Perdew-Burke-Ernzerhof (PBE) parametrization is used for the exchange-correlation energy (Perdew et al., 1996). A supercell for Cu with sizes of $3.48\,\mathrm{nm} \times 0.44\,\mathrm{nm} \times 1.90\,\mathrm{nm}$ containing an edge dislocation is prepared by structural relaxation using EAM. A cutoff energy of $500\,\mathrm{eV}$ is chosen for the plane waves and a $1 \times 5 \times 1$ ($k_x \times k_y \times k_z$) Monkhorst-Pack $\mathbf{k}$-grid is used to sample the Brillouin zone (Monkhorst & Pack, 1976). The convergence of self-consistent field (SCF) calculations using the plane-wave cutoff and $\mathbf{k}$-grid meshing is assured to be below $1\mathrm{meV/atom}$. Similar to the setup in MD simulations, a step-wise strain increment of $4 \times 10^{-3}$ is applied. The Peierls stress is calculated as the minimum stress at which the dislocation is activated to move.

## A.3    MLFF CALCULATIONS

The neuroevolution-potential (NEP) framework is adopted to develop MLFFs for fcc Cu, Al, bcc Fe, and hcp Ti (Fan et al., 2021; Song et al., 2023). The local atomic environments are encoded by

two-body (radial) and three-body (angular) descriptors. A FNN with one hidden layer (30 neurons) is used to predict atomic energies from these descriptors. For systems considering dislocation motion and plastic flow, configurations with applied strain and random perturbation of atomic positions, surfaces, and stacking faults are included in the training set, and the energies of these configurations are labeled using DFT calculations. Instead of using gradient descent-based back-propagation to update the parameters of neural networks, the separable natural evolution strategy algorithm is implemented in the training process to minimize the relatively complex loss functions (Fan et al., 2021; Song et al., 2023). The well-trained MLFFs achieve a prediction accuracy of $< 1$ meV/atom in the energy of atoms and $< 50$ meV/Å in the force on atoms. Atomic simulations using the well-trained MLFFs accurately predict the $\gamma$ surfaces, comparable to the DFT calculations but with three orders lower computational cost.

## A.4 MODELING MORPHOLOGICAL DEVELOPMENT

Brain development is regulated by genetic, molecular, cellular, and mechanical factors across multiple spatiotemporal scales (Klingler et al., 2021; Llinares-Benadero & Borrell, 2019b), and the differential tangential growth hypothesis is commonly used (Tallinen et al., 2016; Klingler et al., 2021; Llinares-Benadero & Borrell, 2019b). FEA can model morphological evolution during brain growth at the continuum level (Tallinen et al., 2016; 2014; Darayi et al., 2022; Budday & Steinmann, 2018; Wang et al., 2021). In the TG model, the tangential growth of the outer gray matter is faster than the inner white matter (Tallinen et al., 2016). Compression resulting from the mismatch in deformation may then lead to mechanical instabilities of the brain surface, forming characteristic sulci and gyri structures (Tallinen et al., 2014; 2016; Striedter et al., 2015; Darayi et al., 2022; Wang et al., 2021; Budday & Steinmann, 2018; da Costa Campos et al., 2021).

In continuum modeling, the reference configuration can be mapped to the current one through the deformation gradient tensor as

$$\mathbf{F} = \mathbf{F}^{e} \cdot \mathbf{G}, \tag{6}$$

where $\mathbf{F}^{e}$ is the elastic deformation gradient and $\mathbf{G}$ is the growth term. In the TG model, the growth tensor $\mathbf{G}$ is

$$\mathbf{G} = g\mathbf{I} + (1 - g)\hat{\mathbf{n}} \otimes \hat{\mathbf{n}}, \tag{7}$$

where $\hat{\mathbf{n}}$ is the surface normal of the reference configuration, $\mathbf{I}$ is the unit tensor, and

$$g = 1 + \frac{\alpha_t}{1 + e^{10(\frac{y}{T} - 1)}} \tag{8}$$

is the growth coefficient, where $\alpha_t$ controls the magnitude of local cortical expansion. There is a smooth transition from the surface of the gray matter layer to the white matter layer with a gradually decreasing growth coefficient. $y$ is the distance to the surface, and $T$ is the thickness of the cortex. The brain is modeled as a nonlinear neo-Hookean hyperelastic material, where the strain energy density is

$$W = \frac{\mu}{2}[\text{Tr}(\mathbf{F}^{e}\mathbf{F}^{e\text{T}})J^{-2/3} - 3] + \frac{K}{2}(J - 1)^2, \tag{9}$$

where $\mu$ is the shear modulus, $J$ is the determinant of Jacobian matrix, $K$ is the bulk modulus. For brain growth, a core-shell structure with a spherical geometry is used for its simplicity. The outer radius is 10 mm and the shell thickness ranges from 0.03 to 1.63 mm, which are determined from the measurements of abnormal and normal human cerebral cortices (Fischl & Dale, 2000; Wang et al., 2021). 4-node tetrahedral elements with a density of $10^6$ tetrahedra/cm$^3$ for discretization with the convergence confirmed (Tallinen et al., 2016; Wang et al., 2021). The bulk modulus of the core and shell is 5 times the shear modulus (Tallinen et al., 2016). Following the experimental evidence, the relative shear modulus ($\mu_{\text{shell}}/\mu_{\text{core}}$) ranges from 0.65 to 1 (Budday et al., 2015). The morphogenesis of brains is triggered by internal elastic stresses generated from differential core-shell growth. The interaction between surfaces is modeled with an energy penalty via vertex-triangle contact, which prevents the nodes from penetrating the faces of elements (Ericson, 2004). An explicit solver is used to minimize the total (elastic and contact) energy of the quasi-static system. The time step $\Delta t = 0.05a\sqrt{\rho/K}$ is set to ensure the convergence, where $a$ is mesh size and $\rho$ is mass density (Belytschko et al., 2014).

