# OpenReview forum: "Physics-Transfer Learning: A Framework to Address the Accuracy-Performance Dilemma in Modeling Complexity Problems in Engineering Sciences"
_ICLR.cc/2025/Conference — Submitted to ICLR 2025_

### Official Review · Reviewer_rsNo · 2024-10-25

**Soundness:** 1
**Presentation:** 2
**Contribution:** 1
**Rating:** 1
**Confidence:** 4

**Summary:**

This paper proposes a framework called Physics Transfer Learning (PT) that aims to learn underlying physics from low-fidelity data to enable extrapolation to high-fidelity data. The authors conduct experiments on two entirely different domains: crystal and brain morphologies. While the paper claims to learn the underlying physics through the PT framework, it does not propose any specific method for achieving this beyond simply inputting data and training a model via supervised learning. Furthermore, there is a significant lack of consideration for competing methods and related work in AI, which raises substantial concerns about the contribution and novelty of the work.

**Strengths:**

1. The idea of considering ellipsoids in brain morphology analysis is intriguing. With more extensive analysis and theoretical development, this concept has the potential for significant advancement in the field.

**Weaknesses:**

1. Despite the title "Physics Transfer," the proposed framework does not modify the input data format used in existing Machine Learning Force Field (MLFF) methods or models that embed brain networks. No additional optimization techniques are introduced. The framework does not adequately consider physics principles or employ transfer learning methodologies.

2. The paper lacks any discussion of AI methodologies or competing methods. The absence of comparisons with existing approaches makes it difficult to evaluate the effectiveness and innovation of the proposed framework.

3. There is no illustration or explanation of how the "physics" is learned within the model. The paper fails to demonstrate the underlying mechanisms that enable the model to capture or learn physical laws.

4. The structure and composition of the proposed framework remain unclear. The paper does not present a cohesive framework that can be commonly applied across two entirely different domains, such as crystal structures and brain morphologies.

**Questions:**

1. Model References: Which models did you reference or build upon for your experiments? Specifically, what is the CNN-based crystal model you used, and what Graph Neural Network (GNN) methods were applied? These details are not provided in the paper, and including them would help in understanding and replicating your work.

2. Learning Physics: You claim that your framework learns physics, but how is this achieved? In the context of Physics-Informed Neural Networks (PINNs), learning physics involves incorporating underlying equations to solve partial differential equations (PDEs). Does your approach relate to methods like "PINNsFormer: A Transformer-Based Framework For Physics-Informed Neural Networks" (ICLR 2024, Zhao et al.)?

3. Related Work: Why did you not include recent relevant papers in your results? In the MLFF field, numerous models like SchNet, DimeNet, Allegro, and Equiformer are well-established. Methods utilizing crystal lattice parameters, such as Matformer, PotNet, Crystalformer, and Comformer, were also not considered. How does your work compare to these existing methods, and why were they omitted from your analysis?

4. Domain Combination: What is the rationale behind combining two domains with no apparent commonality, such as crystals and brain morphologies? How does this contribute to the generality or applicability of your proposed framework? A clear explanation would help in understanding the motivation and potential benefits.

5. Clarification of Figure 1(a): Could you clarify the content and purpose of Figure 1(a)? The caption reads: "Machine learning, constrained by data density and coverage, serves as a potent complement to traditional theories for interpolating and extrapolating solutions, especially as data quality and quantity increase." However, it's unclear how this statement is represented in the figure. Additionally, the text refers to Figure 1(a) as illustrating recent advancements in ML and AI as a data-driven alternative. A more detailed explanation would enhance the reader's comprehension of your conceptual framework.

6. Brain Morphology Representation: The shape of the brain is highly complex and significantly different from simple spheres or ellipsoids. Can using such simplified geometric shapes genuinely benefit the model in understanding brain morphology? Handling data from different domains is sensitive and typically requires careful adaptation or alignment processes. How do you justify that combining these datasets without such processes would be beneficial to the model's performance?

7. Transfer Learning Techniques: The term "Physics Transfer" suggests the use of transfer learning methodologies. What specific transfer learning methods did you employ in your framework? Detailing these techniques would strengthen your contribution and clarify how physics is transferred between domains.

---

> ### Author Response · Authors · 2024-11-30
> **Replies to the comments**
>
> Thank you for the kind review and we appreciate the constructive comments that can help us to improve the work.
>
> Summary
> Thank you for the comments. We propose a framework from the application perspective. We did not intend to propose a new NN model here. We will improve our work following your constructive comments. However, we expect a very significant change in our manuscript, and thus we will not submit a revised submission to ICLR 2025.
>
> Weakness
> We are sorry but I did not get your point. Compared to transfer learning, our work is mainly based on the understanding of the mechanics and physics community. We did not transfer the full information of the problem and apply purely statistical learning to it. Instead, we selected the physics-relevant information and used them as the input of the NN. We are not proposing an MLFF model, and we are not clear what is the `optimization' you were mentioning. We explicitly include the anisotropic elastic constants and #\gamma$ surface as the physics inputs in the model, for example, in the alloy study.
>
> Questions
> Our work is irrelevant to CNN-based crystal models, PINNs, and the implementation of MLFFs.
> Our approach differs from previous physics-informed frameworks in the sense that our physics constraints are `weak'. We do not want to make the data strictly fit into a mathematical model. Instead, we identify the key physics via selected parameters and use them to transfer the physics. Importantly, the validity of the approach is supported in principle from the decade-long understanding in the mechanics and physics community, and are improved here by statistical learning and validated by the presented results.
> We will follow your suggestions to split the work into two.

---

### Official Review · Reviewer_81CL · 2024-11-01

**Soundness:** 2
**Presentation:** 2
**Contribution:** 1
**Rating:** 1
**Confidence:** 3

**Summary:**

The authors purport to develop a novel framework for modeling physical systems using deep learning architectures in this work. The authors claim that models that first learn the "physics" of a given simple system will generalize better to more complex system variants. The authors utilize metallic alloy strength and brain morphological development as two systems of study, providing simple examples of how learning lower-fidelity models can aid in modeling more complex systems.

**Strengths:**

Transfer learning from low to high fidelity systems is an interesting sub-domain within transfer learning at large. I thank the authors for highlighting this problem in their work, even if it is a domain which has already been significantly explored elsewhere.

I do think the core intuition at work in this paper is interesting - the use of lower-fidelity models for developing physics is quite common in the engineering sciences and the sciences at large; however, I think this core insight is marred by a lack of detail or acknowledgement of other related works which approach these problems in similar ways.

**Weaknesses:**

I am chiefly concerned with the significance of this contribution to the literature on transfer learning and the deep learning community at large. Many approaches to transfer learning using lower-fidelity systems or simulations exist already in the literature, and it is well-understood that such approaches can provide benefits over training directly on the more complex system. It is not clear to me how the approach in this work differs from these methodologies significantly other than in applications.  See [1], [2], [3], [4] for some examples which utilize a very similar underlying approach to that in this work. I would like to challenge the authors to differentiate their work more from this existing literature and consider a resubmission. If anything, the authors should consider these other works as baseline approaches for the sake of comparison.

Additionally, this paper suffers from a lack of detail regarding the proposed framework. The most obvious omission is any rigorous definition of what the "learning the physics" means within this work. In previous literature, "learning the physics" more frequently means learning a set of differential equations which describe the system, rather than learning a black-box model with parameters which can predict system changes, but which doesn't provide any direct physical interpretation. Based on the discussion in section 2, the authors seem to be utilizing the latter kind of approach. It is not at all clear to me how learning a simple convolutional neural network provides any manner of learning of the physics of a system. The authors either need to clarify this connection, or clarify that they are doing something other than learning the physics of the system. I am willing to reconsider this point given a very compelling argument from the authors; however, I think even if an argument is provided here, a more significant revision would be needed to further clarify the details omitted in this work.

[1] De, Subhayan, et al. "On transfer learning of neural networks using bi-fidelity data for uncertainty propagation." International Journal for Uncertainty Quantification 10.6 (2020).

[2] Chakraborty, Souvik. "Transfer learning based multi-fidelity physics informed deep neural network." Journal of Computational Physics 426 (2021): 109942.

[3] Liu, Zeyu, Meng Jiang, and Tengfei Luo. "Leveraging low-fidelity data to improve machine learning of sparse high-fidelity thermal conductivity data via transfer learning." Materials Today Physics 28 (2022): 100868.

[4] Song, Dong H., and Daniel M. Tartakovsky. "Transfer learning on multifidelity data." Journal of Machine Learning for Modeling and Computing 3.1 (2022).

**Questions:**

See above:
1) how does this framework differ from other existing work in the literature regarding transfer learning from low to high fidelity systems?
2) how exactly does learning a black-box model which can predict the system count as learning the "physics" of the system?

---

> ### Author Response · Authors · 2024-11-30
> **Replies to the comments**
>
> Thank you for the kind review and we appreciate the constructive comments that can help us to improve the work.
>
> Strengths
>
> Compared to transfer learning, our work is mainly based on the understanding of the mechanics and physics community. We did not transfer the full information of the problem and apply purely statistical learning to it. Instead, we selected the physics-relevant information and used them as the input of the NN. We will improve our work following your suggestions. However, we expect a very significant change in our manuscript, and thus we will not submit a revised submission to ICLR 2025.
>
> Weakness
>
> Our approach differs from previous physics-informed frameworks in the sense that our physics constraints are `weak'. We do not want to make the data strictly fit into a mathematical model. Instead, we identify the key physics via selected parameters and use them to transfer the physics. The recommended papers are valuable and we will include them in our discussion.
>
> Questions
> See our replies in the Strengths and Weakness.

---

### Official Review · Reviewer_3MYx · 2024-11-02

**Soundness:** 2
**Presentation:** 3
**Contribution:** 2
**Rating:** 5
**Confidence:** 3

**Summary:**

In this manuscript, the authors present a physics-transfer (PT) learning framework to merge physics-based modeling and machine learning techniques in complex engineering problems. The framework aims to resolve the accuracy-performance dilemma by learning physics across digital models of varying fidelities. The paper demonstrates the framework's capabilities through two case studies: predicting the strength of metallic alloys and modeling the morphological development of human brains. The authors claim that their approach not only enhances predictive accuracy but also provides new insights into the underlying physics of these systems.

**Strengths:**

The PT framework represents a novel integration of physics and machine learning, introducing an approach to the challenges posed by multiscale problems in engineering sciences. By considering both low-fidelity and high-fidelity models, the authors creatively combine existing ideas to form a new methodology. The motivation for the PT framework is articulated clearly, establishing a strong rationale for the research. The paper successfully outlines the accuracy-performance dilemma, making it accessible to readers familiar with the challenges in engineering modeling. Furthermore, the potential applications in materials science and neuroscience could lead to substantial advancements in these fields.

**Weaknesses:**

The manuscript lacks a comprehensive comparison with existing methods such as $\Delta$-learning and transfer learning. A more in-depth analysis highlighting the advantages and limitations of the PT framework relative to these approaches may enhance the credibility of the manuscript. Specific metrics and results demonstrating improved performance would provide stronger evidence of the framework's contributions.
The manuscript contains sections that are overly technical, which may limit its accessibility to a broader audience. E.g., the explanation of the PT framework's mechanics could benefit from simpler language and additional visual aids to enhance understanding.
The manuscript does not adequately address the scalability of the PT framework, particularly regarding its application to larger datasets or more intricate models. A discussion about the challenges and potential solutions for scaling the approach would be valuable.

**Questions:**

1. Can the authors provide additional metrics or results comparing the PT framework's performance to that of existing methods like $\Delta$-learning and LFAF? This would help clarify the unique contributions of the PT approach.
2. What specific strategies do the authors envision for scaling the PT framework to accommodate larger datasets or more complex problems? Insights into this would enhance the practical relevance of the framework.
3. How do the authors guarantee that the learned physics from the PT framework remains interpretable and relevant in practical applications? A discussion on the interpretability of the model's outputs would be beneficial.

---

> ### Author Response · Authors · 2024-11-30
> **Replies to Comments**
>
> Thank you for the kind review and we appreciate the constructive comments that can help us to improve the work.
>
> Strengths
>
> We are grateful for your summary of our work.
>
> Weakness
>
> Your specific comments on the comparison (with $\Delta$ and transfer learning) and presentation of the work are very helpful, and we will improve our work accordingly. However, we expect a very significant change in our manuscript, and thus we will not submit a revised submission to ICLR 2025.
>
> Questions
> 1. The major difference from our framework with $\Delta$ and LFAF is that we add physics to the model by specifying the inputs, and the physics has been validated by previous studies in the mechanics and physics community, although the cost of the traditional approach there is significant as the models become complex (for example, electronic structure level description of interatomic interaction is needed or complex geometries such as the brain morphology are considered).
> 2. For the metallic alloy study, we did this for elemental metals and specific binary alloys but did not explore more.
> 3. We did this by exploration and exploitation. We added traditional understandings from the mechanics and physics community. We also improve them with new inputs from analysis of the statistical learning results if they are necessary.

---

### Official Review · Reviewer_QoBw · 2024-11-03

**Soundness:** 2
**Presentation:** 2
**Contribution:** 2
**Rating:** 5
**Confidence:** 1

**Summary:**

This paper introduces a physics-transfer (PT) learning framework designed to bridge digital models of varying fidelities and complexities, effectively addressing the accuracy-performance trade-off in multiscale problem analysis. By leveraging PT learning, the model achieves reduced computational costs compared to traditional machine learning models, making it a more efficient solution for understanding complex, representative physical phenomena.

**Strengths:**

This model achieves lower computational costs than traditional ML models, with a method that is easy to understand.

**Weaknesses:**

1. No new models have been proposed.
2. Why was CNN chosen over other state-of-the-art models?
3. The comparison methods are insufficient.
4. Appropriate statistical analysis is required.

**Questions:**

How is the credibility of the model evaluated?

---

> ### Author Response · Authors · 2024-11-30
> **Replies to the comments**
>
> Thank you for the kind review and we appreciate the constructive comments that can help us to improve the work.
>
> Strengths
>
> We did not reduce the computational costs of ML models. Instead, we trained the model from a low-fidelity description of interatomic interaction or simple geometries. We used it to make predictions from high-fidelity parameters (a direct calculation at this level is not feasible or too costly).
>
> Weakness
>
> 1. We did not intend to propose a new NN architecture. We want to emphasize the transfer of physics is possible by using simple NN setups, which are supported by the decade-long research on the models in the mechanics and physics community.
> 2. CNN is simple and other NNs are expected to apply as well.
> 3. The results are validated by comparison of the predicted material properties and brain morphologies. We would be grateful if you could add some specific comments to help us improve the work.
> 4. We would be grateful if you could add some specific comments helping us to improve the work (statistics for what).
>
> Questions
>
> See our reply for Weakness #3.

---

### Meta-Review · Area_Chair_YYPC · 2024-12-20

**Metareview:**

The paper presents a transfer learning framework to learn the underlying physics of complex system through low fidelity simulations and transfer this knowledge to generalize to complex systems. The paper applies this framework to two interesting scientific applications and presents results that highlight its performance.

Strengths: The paper tackles an important and challenging research direction in extrapolating physics from low fidelity data
Weaknesses: Insufficient details to judge the contributions, insufficient comparisons to other transfer learning frameworks

**Additional Comments On Reviewer Discussion:**

The reviewers agreed on the significance of this research but were unable to judge the contributions due to lack of technical details regarding the learnt physics, the mechanics of how physics is transferred to the more complex systems, and insufficient comparisons with numerous other works in this domain.

The authors acknowledged that significant revision would make the paper's contributions stronger.

---

### Decision · Program_Chairs · 2025-01-22

Reject